# Comparison of Functional Components and Antioxidant Activity of *Lycium barbarum* L. Fruits from Different Regions in China

**DOI:** 10.3390/molecules24122228

**Published:** 2019-06-14

**Authors:** Youyuan Lu, Sheng Guo, Fang Zhang, Hui Yan, Da-Wei Qian, Han-Qing Wang, Ling Jin, Jin-Ao Duan

**Affiliations:** 1Jiangsu Collaborative Innovation Center of Chinese Medicinal Resources Industrialization, State Administration of Traditional Chinese Medicine Key Laboratory of Chinese Medicinal Resources Recycling Utilization, National and Local Collaborative Engineering Center of Chinese Medicinal Resources Industrialization and Formulae Innovative Medicine, Nanjing University of Chinese Medicine, Nanjing 210023, China; luyylhy@163.com (Y.L.); fangzhang@njucm.edu.cn (F.Z.); glory-yan@163.com (H.Y.); qiandwnj@126.com (D.-W.Q.); 2School of Pharmacy, Ningxia Medical University, Yinchuan 750021, China; wwwhhq@163.com; 3School of Pharmacy, Gansu University of Chinese Medicine, Lanzhou 730000, China; zyxyjl@163.com

**Keywords:** goji, functional components, antioxidant activity, quality characteristic, producing regions

## Abstract

The fruit of *Lycium barbarum* L. (FLB) has been used as medicines and functional foods for more than 2000 years in East Asia. In this study, carotenoid, phenolic, flavonoid, and polysaccharide contents as well as the antioxidant activities of FLB from 13 different regions in China from a total of 78 samples were analyzed. The results showed that total carotenoid contents ranged from 12.93 to 25.35 mg β-carotene equivalents/g DW. Zeaxanthin dipalmitate was the predominant carotenoid (4.260–10.07 mg/g DW) in FLB. The total phenolic, total flavonoid, and total polysaccharide contents ranged from 6.899 to 8.253 mg gallic acid equivalents/g DW, 3.177 to 6.144 mg rutin equivalents/g DW, and 23.62 to 42.45 mg/g DW, respectively. Rutin content ranged from 0.1812 to 0.4391 mg/g DW, and ferulic acid content ranged from 0.0994 to 0.1726 mg/g DW. All of these FLB could be divided into two clusters with PCA analysis, and both individual carotenoids and total carotenoid contents could be used as markers for regional characterization. The phenolic components were the main substance for the antioxidant activity of FLB. Considering the functional component and antioxidant activities, FLB produced in Guyuan of Ningxia was the closest to Daodi herbs (Zhongwei of Ningxia), which is commercially available high quality FLB. The results of this study could provide guidance for comprehensive applications of FLB production in different regions.

## 1. Introduction

Fruit of *Lycium barbarum* L. (FLB), widely known as goji or wolfberry, has been used as a medicine and functional food for more than 2000 years in East Asia, especially in China. It has also been recognized as a ‘super food’ or ‘super fruit’ in Europe and North America [1]. In traditional Chinese medicine, it is believed that FLB could nourish the liver and kidney, and improve eyesight [2,3]. Modern pharmacology research has confirmed that FLB has the properties of antioxidant, antiaging, immune modulation, antidiabetic, etc. [3,4].

Carotenoids, flavonoids, phenylpropanoids, coumarins, lignans, amides, alkaloids, glycerogalactolipids, and polysaccharides build the compound library of FLB [5,6,7,8]. Until now, a total of 11 free carotenoids, such as zeaxanthin, β-cryptoxantin, β-carotene, etc., and 53 polyphenols, such as ethyl *trans*-*p*-coumarate, ethyl *trans*-ferulate, *trans*-sinapinic, syringenin, *cis*-ferulic acid, etc., were isolated from FLB [8,9]. Previous studies showed that carotenoids from FLB could enhance immune system functioning, prevent chronic diseases, and possess biological activities, such as antiaging, antidiabetic, anticancer, and so on [10,11,12,13]. Phenolic compounds from FLB show antioxidant, antiaging, anti-inflammatory, antihyperglycemic, etc. [14,15,16]. Furthermore, FLB polysaccharides exhibit a variety of functions, including antiaging, antitumor, antidiabetic, immunomodulation, hypoglycemic, and hypolipidemic activities [12,17,18,19]. Thus, carotenoids, phenolic compounds, and polysaccharides in FLB have received increasing attention for their health promoting properties [11,20,21].

In recent years, the demand for FLB has increased due to its potential beneficial effects on human health [21]. In 2017, the output of FLB in China exceeded 265,000 tons, and the export volume was 12,200 tons [22]. To meet the market demand, *L. barbarum* has been widely cultivated in northwest China, such as Ningxia, Gansu, Inner Mongolia, Qinghai, and Xinjiang. Daodi herbs, produced in specific geographic regions (with designated ecological environment and appropriate cultivation, harvesting, and processing technique), are widely recognized as a symbol of good quality with better beneficial clinical effects than those of non-Daodi herbs produced in other regions [23]. So far, reports about the bioactive components of FLB from different regions are limited. Although there are reports that the molecular weight distributions and potential health effects are similar for polysaccharides in FLB collected from different regions of China [24,25], there are a lack of studies on whether the profile of other functional constituents, such as carotenoids, phenolic, flavonoid, etc., and the antioxidant activities of FLB produced in different regions are different, and an identification of regions that have the potential to produce high quality FLB with a similar functional composition to those of high quality Daodi herbs.

In the present study, carotenoid, flavonoid, phenolic, and polysaccharide contents in FLB produced in different regions were determined and compared, and their antioxidant activities were evaluated. These findings provide useful information for illustrating the quality of FLB to ensure the efficiency and utilization of FLB from different regions.

## 2. Results and Discussion

### 2.1. Carotenoid Content in Fruits of L. barbarum

#### 2.1.1. Optimization of Extract Preparation

Carotenoids are conventionally obtained with non-polar solvents (e.g., hexane, petroleum ether, and tetrahydrofuran) for the extraction of non-polar carotenes or esterified xanthophyll, and polar solvents (e.g., acetone, ethanol, ethyl acetate) for the extraction of polar carotenoids [26]. Meanwhile, to prevent enzymatic oxidation, a short extraction time with an appropriate temperature is recommended for effective extraction [26]. Thus, both non-polar and polar solvents (hexane/acetone/ethanol) were used to obtain the maximum extraction for carotenoids in FLB based on the method described by Liu et al. [27]. In addition, the extraction time (20, 30, 40, 50, and 60 min), extraction temperature (20, 30, 40, 50, and 60 °C), and sample:solvent ratios (1:40, 1:60, 1:80, 1:100, and 1:120) were investigated. The results are shown in Appendix A. It revealed that an ultrasonic bath extraction with 0.5 g of fruit powder mixed with hexane/acetone/ethanol (2:1:1, *v*/*v*/*v*) in the 1:80 sample:solvent ratio at 30 °C for 40 min were the optimum conditions for the extraction of individual carotenoids (extracts I) in FLB.

#### 2.1.2. Development and Validation of the HPLC Method for Carotenoid Analysis

It has been reported that reversed phase chromatography with a C_30_ column obtains better separation for carotenoids [28,29]. In this study, a YMC HPLC C_30_ column (150 × 4.6 mm, 3μm) with mobile phases A (methanol/methyl t-butyl ether (MTBE)/water, 81:15:4, *v*/*v*/*v*) and B (methanol/MTBE/water, 16:80:4, *v*/*v*/*v*) was used to separate the carotenoids based on the method of Petry and Mercadante [30]. A gradient elution was used and the best separation was obtained in 40 min (Figure 1). The established HPLC method for the quantitative analysis of carotenoids in FLB was validated for its linearity, limits of detection (LOD), limits of qualification (LQD), intra-day and inter-day precisions, repeatability, and accuracy. The results are shown in Appendix A. All calibration curves (*r^2^* > 0.9950) indicated a good correlation between the compound concentrations and the peak areas within the test ranges. The LODs and LOQs were in the range of 0.0242 to 0.0595 μg/mL and 0.0847 to 0.2021 μg/mL, respectively. The precision, repeatability, and stability variation (RSD) were all less than 2.0%. The recoveries were between 95.7% and 98.4% with an RSD of less than 3.0%. All the results indicated that the method was repeatable and accurate according to the International Conference on Harmonization (ICH) guideline [31].

#### 2.1.3. Analysis of Carotenoids in FLB from Different Regions

The results of carotenoid contents in 78 samples of FLB are shown in Appendix A, which are also summarized in Table 1 based on the 13 main production regions. The total carotenoid content (TCC) analyzed by spectrophotometry in the FLB ranged from 12.93 to 25.35 mg β-carotene equivalents (βCE)/g dry weight (DW). The TCC in the FLB from different regions, from high to low was: Inner Mogolia (IMWL) > Baiyin of Gansu (GSBY) > Guyuan of Ningxia (NXGY) > Zhangye of Gansu (GSZY) > Huinong of Ningxia (NXHN) > Ge’ermu of Qinghai (QHGE) > Wuyuan of Inner Mongolia (IMWY) > Yinchuan of Ningxia (NXYC) > Zhongwei of Ningxia (NXZW) > Delingha of Qinghai (QHDL) > Hanggin Rear Banner of Inner Mongolia (IMHJ) > Jiuquan of Gansu (GSJQ) > Jinghe of Xinjiang (XJJH). Zhang et al. reported that the TCC in the fresh fruits of *L. barbarum* L. ranged from 7.97 to 11.33 mg βCE/g [5]. The TCC reported by Zhang et al. [5] was higher than the results in this study, however, given that the moisture content in fresh samples of FLB is 70% to 75%, this may be caused by the different states of the fruits (fresh and dry) and the different extraction processes.

For the individual carotenoids analyzed by HPLC, zeaxanthin dipalmitate was the highest and its content ranged from 4.260 to 10.07 mg/g DW, accounting for 32.94% to 45.55% of the total carotenoid, followed by zeaxanthin (0.0277–0.2857 mg/g DW), β-carotene (0.0229–0.1301 mg/g DW), and β-cryptoxanthin (0.0001–0.0031 mg/g DW). Meanwhile, lutein and neoxanthin were not detected in all samples. ANOVA analysis showed that the individual carotenoid contents in FLB from different regions was significantly different (*p* < 0.01). The individual carotenoid contents in samples from IMWL were the highest, followed by the samples from GSBY. However, no significant difference was found for the contents of β-carotene and zeaxanthin dipalmitate in FLBs from IMWL and GSBY. In addition, the contents of zeaxanthin dipalmitate in samples produced in NXHN and NXGY, GSZY and QHGE were also not significantly different. The individual carotenoid content in FLB from Jinghe of Xinjiang (XJJH) was the lowest. There was no significant difference in the zeaxanthin dipalmitate contents between the samples produced in IMHJ and GSJQ. In addition, it was notable that a significant difference existed between the sum of individual carotenoids (4.326–10.49 mg/g DW) and the TCC (12.93–23.35 mg βCE/g DW). According to the literature [9], an assumption was used stating that some carotenoids exist in FLB besides the six compounds analyzed in this assay. The more important reason for the significant difference is that UV spectrophotometry has poor selectivity in quantitative analysis, due to the presence of other chemical compounds that can absorb electromagnetic radiation [32].

Inbaraj et al. [9] identified 11 free carotenoids (including five zeaxanthin isomers, four β-carotene isomers, neoxanthin, and β-cryptoxanthin) and seven carotenoids esters (including three zeaxanthin monopalmitate isomers, three β-cryptoxanthin monopalmitate isomers, and one zeaxanthin dipalmitate) from FLB extract by HPLC-DAD-APCI-MS, and the results showed that carotenoid esters had the largest amount, especially zeaxanthin dipalmitate. In this study, the carotenoid contents in FLB from the main cultivated regions of China were determined. The results also showed that zeaxanthin dipalmitate was the predominant component in the carotenoids of FLB. Additionally, Inbaraj et al. [9] and Zhang et al. [5] both reported that neoxathin was detected in FLB extract after a saponification process. However, this compound was not detected in all samples without the saponification process in this study. Owing to the characteristics of carotenoids, it was suggested that neoxanthin in saponified extract may be transformed from other components during the process of saponification. Zhang et al. [5] reported that lutein was found in FLB, while it was not detected in the samples of this study.

Traditional and clinical experience hold that the curative effect of Daodi herbs is better than non-Daodi herbs in traditional Chinese medicine [23], and the curative effect is based on the active constituents. Thus, the carotenoid contents in FLB from non-Daodi herbs regions and Daodi herbs region were compared. As Table 1 shows, the total carotenoid, β-carotene, β-cryptoxanthin, and zeaxanthin dipalmitate contents of FLB produced in IMWL and GSBY were significantly higher (*p* < 0.01) than those produced in NXZW (the Daodi herbs regions of FLB), and those produced in XJJH were significantly lower than those produced in NXZW. However, for the zeaxanthin content, all the samples except those from NXGY exhibited a significant difference with those from NXZW (*p* < 0.05). Thus, considering the total and individual carotenoid contents, FLB produced in NXGY was the closest to Daodi herbs (FLB produced in NXZW) due to their similar carotenoid profiles.

### 2.2. Total Phenolic Content and Flavonoid Content in FLB

#### 2.2.1. Optimization of Extract Preparation

Phenolic acids and flavonoids are always extracted by different concentrations of methanol or ethanol solution [33,34]. However, high concentrations of carotenoids in FLB could lead to the extract exhibiting a faint yellow color. To ascertain whether the color of the extract affected the analysis results of the phenolic, two extraction methods were compared: (1) The optimized method used to extract carotenoids in this study was performed to remove carotenoids, then the extracted residues were dried and suspended in 80% ethanol solutions to extract phenolic compounds using an ultrasonic bath; and (2) 80% ethanol solution was used to extract phenolic compounds with an ultrasonic bath directly. The results showed that the phenolic compound contents did not exhibit a significant difference between the two methods, which illustrated that the faint yellow color of the ethanol extract does not affect the analysis of phenolic compounds. Thus, the 80% ethanol solution was used to extract phenolic compounds with an ultrasonic bath directly. In addition, different concentrations of ethanol solution (90%, 80%, 70%, 60%, and 50%), extraction time (40, 60, 80, 100, and 120 min), extraction temperature (20, 30, 40, 50, and 60 °C), and sample:solvent ratios (1:40, 1:60, 1:80, 1:100, and 1:120 *w*/*v*) were investigated. The results are shown in Appendix A. It revealed that the ultrasonic bath extraction with 0.5 g of fruit powder soaked for 30 min with 80% ethanol solutions in a 1:80 sample:solvent ratio at 50 °C for 40 min was the final optimized conditions for the extraction of phenolic compounds (extract II) in FLB.

#### 2.2.2. Analysis of Phenolic and Flavonoid Contents in FLB from Different Regions

The results of the phenolic and flavonoid contents in the 78 samples of FLB are shown in Appendix A, which are also summarized in Table 1 based on the 13 main production regions. The total phenolic content (TPC) of FLB ranged from 6.899 to 8.253 mg gallic acid equivalents (GAE)/g DW, and that from NXZW was the highest, followed by GSZY (7.966 mg GAE/g DW) and NXHN (7.902 mg GAE/g DW), while no significant difference was found between them (*p* > 0.05). The samples from IMHJ (6.899 mg GAE/g DW) and QHGE (6.960 mg GAE/g DW) were the lowest. The TPC in FLB from different regions, from high to low was: NXZW > GSZY > NXHN > QHDL > NXYC > GSBY > NXGY > GSJQ > XJJH > IMWY > IMWL > QHGE > IMHJ. Compared with Daodi herbs (NXZW), the TPCs in FLB from GSJQ, GSBY, IMHJ, IMWY, IMWL, QHGE, and XJJH were significantly lower (*p* < 0.05). The study results illustrated that the TPC in FLB varied with the produced regions. The TPC in FLB was consistent with the report by Zhao et al. [35], and was higher than the new goji cultivars which are produced in Poland [15].

The total flavonoid content (TFC) of FLB from different regions varied, and ranged from 3.177 to 6.144 mg rutin equivalents (RE)/g DW, which is lower than the report by Zhao et al. [35]. The samples from GSBY exhibited the highest TFC content, followed by IMWY (5.977 mg RE/g DW), NXGY (5.936 mg RE/g DW), GSZY (5.821 mg RE/g DW), and NXZW (5.545 mg RE/g DW), while no significant difference was found among them (*p* > 0.05). The TFC in samples from XJJH was the lowest. Generally, the TFC in FLB from different regions, from high to low was: GSBY > IMWY > NXGY > GSZY > NXZW > NXHN > IMWL > GSJQ > NXYC > IMHJ > QHDL > QHGE > XJJH. Compared with Daodi herbs (produced in NXZW), those produced in NXYC, GSJQ, GSBY, IMHJ, IMWL, QHDL, QHGE, and XJJH exhibited significant differences in their TFC (*p* < 0.05). For the above two types of components, FLB produced in NXGY, NXHN, and GSZY were not significantly different (*p* > 0.05) with NXZW, and these samples were considered to the closest to Daodi herbs.

Phenolic contents were affected by many factors, such as geographical, environmental, cultivation method, dry method, and so on [5,35,36]. According to the literature [5], we tried to analyze the individual phenolic composition of FLB by HPLC. However, only ferulic acid and rutin could be identified. The HPLC chromatograms of standards and samples of FLB are shown in Appendix A. The results of the method validation are shown in Appendix A. The results of the rutin and ferulic acid contents in 78 samples of FLB are shown in Appendix A, which are also summarized in Table 1 based on the 13 main production regions. Rutin contents varied from 0.1812 to 0.4391 mg/g DW, and ferulic acid contents ranged from 0.0994 to 0.1726 mg/g DW. The rutin content in samples from GSBY was the highest, followed by samples from NXGY. The ferulic acid content in FLB from different regions, from high to low was: GSBY > NXZW > IMWL > NXHN > GSZY > NXGY > XJJH > NXYC > GSJQ > QHDL > IMHJ > QHGE. Zhang et al. [5] reported that rutin and ferulic acid contents in FLB were 67.9 μg/g FW and 152.2 μg/g FW, respectively, which is much higher than the results in this study, given that the moisture content in fresh samples of FLB was 70% to 75%. In addition, the results in this study are also consistent with the reported individual phenolic acid and flavonoid contents in dried samples of FLB being as low as the microgram level [37].

The samples in this study were collected from cultivated regions with exact information, which could represent the quality of FLB originating geographical origins. The study results showed that both the TPC and TFC in FLB from NXZW were higher, which may be correlated with its better clinical effect. Additionally, the function of FLB produced in NXGY, NXHN, and GSZY may be similar with those in NXZW due to their similarity on TPC and TFC.

### 2.3. Analysis of Total Polysaccharide Content (TLBP) in FLB from Different Regions

Polysaccharides from FLB have received extensive attention for their potential health-promoting effects [19,38,39,40]. A typical procedure [19] was followed to prepare the polysaccharide in FLB and hot water was used as the extraction solvent. The results of the TLBP in 78 samples of FLB are shown in Appendix A, which are also summarized in Table 1 based on the 13 main production regions. The TLBP of FLB ranged from 23.62 to 42.45 mg/g DW. The samples from NXHN exhibited the highest polysaccharide content, followed by NXYC, GSBY, and NXZW, and that from QHGEM was the lowest. The TLBP of FLB produced in different regions, from high to low was: NXHN > NXYC > GSBY > NXZW > IMWL > GSJQ > GSZY > NXGY > QHDL > XJJH > IMHJ > IMWY > QHGE. The study results are consistent with the report of Wu et al. [24]. The TLBP of FLB produced in Ningxia, Gansu, Qinghai, Inner Mongolia, and Xinjiang ranged from 22.89 to 45.16, 25.63 to 37.97, 20.55 to 33.62, 21.46 to 39.00, and 22.47 to 33.43 mg/g DW, respectively. Moreover, the TLBPs of FLB from different regions in Gansu, Xinjiang, and Qinghai were consistently higher. However, the consistency of TLBP in FLB from different regions of Ningxia was low, especially the TLBP in FLB from NXHN, which was significantly different from that in other regions of Ningxia. The results showed that the cultivated germchit of FLB all originated from Ningxia. Thus, variation of the TLBP in FLB from different regions may be caused by the environment, producing, and processing method.

Due to the results that the samples from NXZW exhibited no significant difference in TLBP with those from NXYC, NXGY, GSJQ, GSBY, GSZY, IMWL, and QHDL (*p* > 0.05), thus, only considering TLBP, the samples from these regions were considered to be the closest to Daodi herbs, and may have a similar clinical effect.

### 2.4. Antioxidant Activities of FLB from Different Regions

The results of the antioxidant activities for the 78 samples of FLB extracted with extract I (hexane/acetone/ethanol (2:1:1, *v*/*v*/*v*)) and extract II (80% ethanol solution) are shown in Appendix A, which are also summarized in Figure 2 based on the 13 main production regions. The EC_50_ of 1,1-diphenyl-2-picrylhydrazyl free radical (DPPH) for extract I and II ranged from 16.98 to 26.12 mg sample/mL and 2.278 to 3.507 mg sample/mL, respectively. The EC_50_ of 2,2′-azino-bis(3-ethylbenzothiazoline-6-sultonic acid) diammonium salt (ABTS) for extract I and II ranged from 2.557 to 0.057 mg sample/mL and 0.4145 to 0.5046 mg sample/mL, respectively. The antioxidant activity of the ferric reducing/antioxidant potential (FRAP) for extract I and II ranged from 11.45 to 17.05 μmol Fe^2+^/g DW and 74.29 to 94.75 μmol Fe^2+^/g DW. It is worth noting that the antioxidant activities of extract II were 5 to 13 times those of extract I, so the components in extract II contributed the most to the antioxidant activity for FLB.

ANOVA analysis showed that the antioxidant activities of these two extracts of FLB from different regions mostly exhibited a significant difference (*p* < 0.01) with the assays of DPPH, ABTS, and FRAP, respectively. The antioxidant activities between the non-Daodi herb regions and Daodi herb regions of the two extracts showed that the samples from NXYC and NXGY were not significantly different (*p* > 0.05) with that from NXZW (Daodi herb region) for the DPPH assay. For the results of the ABTS assay, the samples from GSZY and NXGY were similar to that from NXZW. In addition, the samples from GSBY, GSZY, and NXGY exhibited no significant difference (*p* > 0.05) with that from NXZW for the FRAP assay. When all the antioxidant activities obtained by the three methods were considered, the samples produced in NXGY were the closest to Daodi herbs (produced in NXZW).

### 2.5. The Regional Characterization of Functional Components in FLB

To obtain the characteristics of carotenoids in FLB produced in different regions, a principal component analysis (PCA) test was performed, and the result is shown in Figure 3A. The first two principal components explained 90.6% (PC1 represents 70% and PC2 represents 20.6%) of the total variation based on the five variables of carotenoids. The PCA score plot separated the samples into two clusters. Cluster I mainly contained samples from IMWL and cluster II contained samples from other regions. The reason that caused the carotenoid contents to be different needs to be further studied.

The PCA test of TPC, TFC, rutin, ferulic acid, and TLBP was performed and the result is shown in Figure 3B. The first two principal components explained 66.0% (PC1 represents 45.1% and PC2 represents 20.9%) of the total variation among the analyzed samples, and the results could not identify the representative regions for those samples with the above three functional components.

When TPC, TFC, rutin, ferulic acid, TLBP, TCC, and individual carotenoids were considered, the samples could be separated into two clusters (I, IMWL; II, other regions) (Figure 3C), which was consistent with the PCA result obtained for the carotenoids. Thus, the carotenoids could be used as markers for the regional characterization of FLB.

## 3. Materials and Methods 

### 3.1. Plant Materials

A total of 78 batches of mature fruits of *L. barbarum* were collected from the 13 production regions of Ningxia, Gansu, Inner Mongolia, Qinghai, and Xinjiang provinces (NXZW, NXHN, NXYC, NXGY, GSBY, GSZY, GSJQ, QHGE, QHDL, IMWL, IMWY, IMHJ, XJJH, Figure 4a) from July to August 2017, and the appearance characteristics of the fruits are shown in Figure 4b. The sample’s information is summarized in Appendix A. The samples were collected from the production regions directly. In order to ensure the reliability of the sample collection, three biological replicates were collected with 1000 g for each batch, and at least 3 sets of samples were collected in the same regions. All the samples were authenticated by Prof. Jin-ao Duan, Nanjing University of Chinese Medicine, Nanjing, China. After collection, the fruits were dried at 60 °C until a constant weight, and ground into a fine powder. The moisture content of the dried samples was determined by a ADAM PMB-53 Moisture Analyzer (cAdam Equipment Inc., Oxiford, UK).

### 3.2. Reagents and Standards

HPLC-grade methyl alcohol and methyl tert-butyl ether (MTBE) were obtained from TEDIA (San Francisco, OH, USA). Deionized water was prepared by a Milli-Q water purification system (Millipore, Billerica, MA, USA). DPPH, ABTS, and 2,4,6-tris (2-pyridyl)-s-triazine (TPTZ) were acquired from Sigma (St. Louis, MO, USA). Other analytical grade reagents, such as hexane and acetone, were supplied by Sino Pharm Chemical Reagent Co., Ltd. (Shanghai, China). Reference compounds of zeaxanthin, lutein, β-carotene, β-cryptoxanthin, and neoxanthin were provided by Sigma (St. Louis, MO, USA), and zeaxanthin dipalmitate was provided by EXTRASYNTHASE (Genay Cedex, France). The purities of zeaxanthin, lutein, β-carotene, β-cryptoxanthin, and neoxanthin were determined to be over 97% and zeaxanthin dipalmitate to be over 95% by HPLC detection. Their structures are shown in Appendix A. Rutin, gallic acid, and glucose were purchased from Beijing Solarbio & Technology Co. Ltd. (Beijing, China) with a purity over 98%.

### 3.3. Analysis of Total Carotenoid Content and Individual Carotenoids

#### 3.3.1. Preparation of Standard Solutions 

Stock solutions of the standards were prepared by dissolving certain amounts of individual compounds in acetone, with a concentration of 0.2 mg/mL for zeaxanthin, lutein, β-carotene, β-cryptoflavin, and neoxanthin, and 1 mg/mL for zeaxanthin dipalmitate. Subsequently, series of diluted standard solutions were prepared in the following concentration ranges for HPLC-DAD analysis: Neoxanthin, 0.25–8.00 μg/mL; lutein, 0.76–12.15 μg/mL; zeaxanthin, 0.48–30.60 μg/mL; β-carotene, 1.12–35.84 μg/mL; β-cryptoxanthin, 0.38–3.00 μg/mL; zeaxanthin dipalmitate, 38.63–618.00 μg/mL. For spectrophotometric analysis of carotenoid contents, a series dilutions of β-carotene solutions were prepared in the concentration range of 2.13–42.47 μg/mL. All the above solutions were stored at 4 °C until use (less than three days), and then filtered through a 0.45 μm nylon membrane provided by Tianjin jinteng experimental equipment Co. Ltd. (Tianjin, China) for HPLC-DAD analysis.

#### 3.3.2. Preparation of Extract I for the Analysis of Carotenoids

The carotenoids were extracted according to the method described by Liu et al. [27] with some modifications. The sample from NXZW (no. 5) was used to optimize the preparation of the sample solution (extract I). According to the single factor experiment design, the extraction time (20, 30, 40, 50, and 60 min), extraction temperature (20, 30, 40, 50, and 60 °C), and sample:solvent ratio (1:40, 1:60, 1:80, 1:100, and 1:120) were investigated with three replicates for each experiment. The individual carotenoid contents were used to evaluate the extraction method. At last, the fruit powder (0.5 g) was extracted with 40 mL of hexane/acetone/ethanol (2:1:1, *v*/*v*/*v*) for 40 min at 30 °C using an ultrasonic bath (40 kHz, Nanjing, China). The extract solutions were centrifuged at 8000 rpm for 10 min, and then the supernatants were dried in a Vacufuge Plus vacuum concentrator (Labconco, Kansas, MO, USA). Acetone/MTBE (3:1, *v*/*v*, 3 mL) was added and filtered through a 0.45 μm nylon membrane for the HPLC analysis.

#### 3.3.3. Analysis of the Total Carotenoid Content (TCC) by Spectrophotometry

TCC content was measured by absorbance at 450 nm using a Philes ultraviolet spectrophotometer (Nanjing Philes Inc., Nanjing, China), and the results are expressed as mg β-carotene equivalents (βCE)/g dry weight (DW) [5].

#### 3.3.4. Analysis of Individual Carotenoids by HPLC-DAD 

HPLC-DAD analysis was performed on a Waters Alliance 2695 system (Waters, Milford, MA, USA) consisting of a 2695 module and a 2998 DAD detector. A YMC HPLC C30 column (150 × 4.6 mm, 3 μm, YMC Co., LTD., Kyoto, Japan) was applied for the separation. The mobile phase was composed of A (methanol/MTBE/water, 81:15:4, *v*/*v*/*v*) and B (methanol/MTBE/water, 16:80:4, *v*/*v*/*v*) with the following gradient elution: 0–7 min, 80%–50% A; 7–19 min, 50%–22% A; 19–22 min, 22% A; 22–37 min, 22%–0% A; 37–40 min, 0%–80% A. The column temperature was set at 25 °C. The flow rate was set at 0.4 mL/min. The injection volume was 10 μL, and the detection wavelength was 450 nm.

#### 3.3.5. Method Validation for the Analysis of Individual Carotenoids

The method validation was carried out following the International Conference on Harmonization (ICH) guideline [31]. The calibration curve for each compound was constructed by plotting the peak areas against the corresponding concentrations. The LOD and LOQ were calculated based on a signal-to-noise of S/N ≥ 3 and S/N ≥ 10, respectively.

For the precision study, the intra-day and inter-day variability were investigated by determining the standard solutions for six replicates during a single day and by duplicating the experiments for three consecutive days.

For the repeatability study, six independent sample solutions prepared from sample no. 5 of NXZW, the information of which is listed in Appendix A, were determined and evaluated.

One of the sample solutions for the repeatability was stored at 10 °C, and was injected into the apparatus at 0, 2, 4, 8, 12, and 24 h, respectively, to evaluate the stability. 

All variations (expressed by the percentage of relative standards deviations) of the peak areas and individual carotenoid contents were taken as the measures of the precision, repeatability, and stability, respectively.

For the accuracy study, the recovery test was performed by adding three different levels (high, middle, and low) of the analyzed compounds to a 0.25 g sample from NXZW for extraction and HPLC analysis. The average recovery of each carotenoid was calculated based on the equation: Recovery (%) = [(amount found − original amount)/amount added] × 100%.

### 3.4. Analysis of Total Phenolic and Total Flavonoid Contents 

#### 3.4.1. Preparation of Standard Solutions

Stock solutions of the standards were prepared by dissolving certain amounts of individual compounds in 80% ethanol, rutin, and gallic acid at a concentration of 2.160 mg/mL and 2.120 mg/mL, respectively. A series of rutin were prepared in the concentration at a range of 0.2880 to 0.8640 mg/mL, and gallic acid of the concentration range of 0.2829 to 0.9902 mg/mL.

#### 3.4.2. Preparation of Extract II for the Analysis of Total Phenolic and Total Flavonoids

Firstly, two extraction methods were compared to identify whether the color of the extract affected the determination results of the total phenolic. At last, 80% ethanol solution was used to extract phenolic compounds with an ultrasonic bath directly for further study. Then, according to the single factor experiment design, the extraction method parameters, such as different concentrations of ethanol solution (50%, 60%, 70%, 80%, and 90%), extraction time (20, 30, 40, 50, and 60 min), extraction temperature (20, 30, 40, 50, and 60 °C), and sample:solvent ratios (1:40, 1:60, 1:80, 1:100, and 1:120), were investigated to optimize the sample solutions’ preparation. At last, the fruit powder (0.5 g) was soaked with 40 mL of 80% ethanol for 30 min and extracted for 40 min at 50 °C using an ultrasonic bath (40 kHz, Nanjing, China). The extract solutions were centrifuged at 8000 rpm for 10 min, and then the supernatants were dried in a Vacufuge Plus vacuum concentrator (Labconco, USA), and then adding 4 mL of 80% ethanol to dissolve the solid for total phenolic, total flavonoid, and antioxidant activity analysis.

#### 3.4.3. Analysis of TPC and TFC by Spectrophotometry

TPC was determined using a Philes ultraviolet spectrophotometer (Nanjing, China) according to the Folin–Ciocalteau method [41], and the results are expressed as mg gallic acid equivalents (GAE)/g DW. TFC was determined based on the method described by Wang et al. [41] with some modifications. The results are expressed as mg rutin equivalents (RE)/g DW.

### 3.5. Analysis of Polysaccharide Contents

Stock solutions of the standards were prepared by dissolving glucose in distilled water, at a concentration of 0.2135 mg/mL. In the UV analysis, a series of glucose was prepared in the concentration range of 0.0213 to 0.1066 mg/mL.

Polysaccharide was extracted based on the method described by Pharmacopoeia of People’s Republic of China [42] with some modifications. Briefly, the fruit powder (0.5 g) was soaked with 40 mL of ether for 20 min and extracted for 60 min using an ultrasonic bath. The supernatant was removed and the residues were then dried, suspended in 20 mL of 80% ethanol, and extracted for 60 min with an ultrasonic bath. Subsequently, the supernatant was also removed, and the residues were dried, suspended in 50 mL of distilled water, and extracted for 120 min with a boiling water bath and 60 min with an ultrasonic bath. Then, the extract was centrifuged at 4000 rpm for 10 min, and the supernatant was collected for analysis.

TLBP was determined according to Pharmacopoeia of People’s Republic of China [42] with some modifications. An aliquot of 0.5 mL of glucose standard or samples solution was added to 1.5 mL of distilled water and 1.0 mL of 5% phenol solution, mixed thoroughly, then 5.0 mL of concentrated sulphuric acid was added. After keeping it for 10 min at room temperature and in a water bath (40 °C) for 15 min, the absorbance was measured at 490 nm using a Philes ultraviolet spectrophotometer (Nanjing, China). The results are expressed as mg/g DW of glucose equivalents.

### 3.6. Antioxidant Activity Assays

For the DPPH assay, the method was carried out according to that of Brand-Williams et al. [43]. The antioxidant activity is expressed as the amount of antioxidant necessary to decrease the initial DPPH concentration by 50% (EC_50_).

For the ABTS assay, the method was performed according to that of Re et al. [44]. Five different concentration sample solutions and a ABTS working solution were added into wells of a 96-well plate, respectively, and the absorbance was detected at 734 nm after 6 min. The antioxidant activity is expressed as EC_50_.

For the FRAP assay, the method was performed according to that of Benzie and Strain [45]. Samples and the FRAP working solution were diluted directly into wells of a 96-well plate, and the absorbance was measured at 593 nm. The activity is expressed as μmol Fe^2+^/g DW.

Extract I and extract II as prepared in Section 3.3.2 and Section 3.4.2, respectively, were used for all antioxidant activity assays.

### 3.7. Data Analysis

All data are expressed as the means ± standard deviation. Statistical analysis was performed using SPSS 19.0 (SPSS Inc., Chicago, IL, USA). Significant differences among different regions were calculated using a one-way ANOVA test, along with Duncan’s test. Meanwhile, the t-test was applied to compare the difference between Daodi herbs and non-Daodi herbs. Principal component analysis (PCA) was performed using SIMCA-P 14.1 (Umetrics Inc., Malmö, Sweden).

## 4. Conclusions

In the present work, the functional components and antioxidant activities of different solvents extracts of FLB from different regions were determined and analyzed. It was found that the functional component contents and antioxidant activities of FLB produced in different regions were significantly different (*p* < 0.01), and were classified into two clusters (IMWL and other regions). Total carotenoid and individual carotenoids represented the regional characterization of FLB. The components of extract I (80% ethanol solution extraction) were the main substance for the antioxidant activity of FLB, especially for TPC. Considering the functional component contents and antioxidant activities, FLB produced in NXGY was the closest to Daodi herbs (NXZW). The results can provide guidance for comprehensive applications of FLB production in different regions.

## Figures and Tables

**Figure 1 molecules-24-02228-f001:**
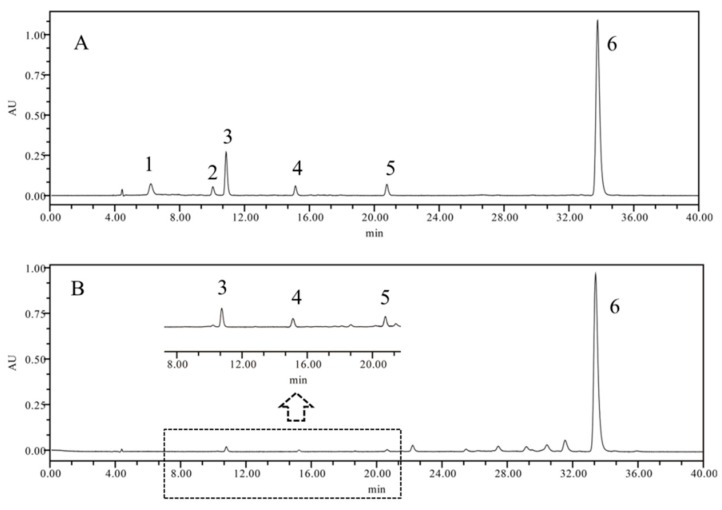
HPLC chromatogram of carotenoids in fruits of *Lycium barbarum* ((**A**) chromatogram of mixed standards solution; (**B**) chromatogram of sample solution (sample no. 5 from Zhongwei of Ningxia); **1.** neoxanthin, **2.** lutein, **3.** zeaxanthin, **4.** β-cryptoflavin, **5.** β-carotene, **6.** zeaxanthin dipalmitate).

**Figure 2 molecules-24-02228-f002:**
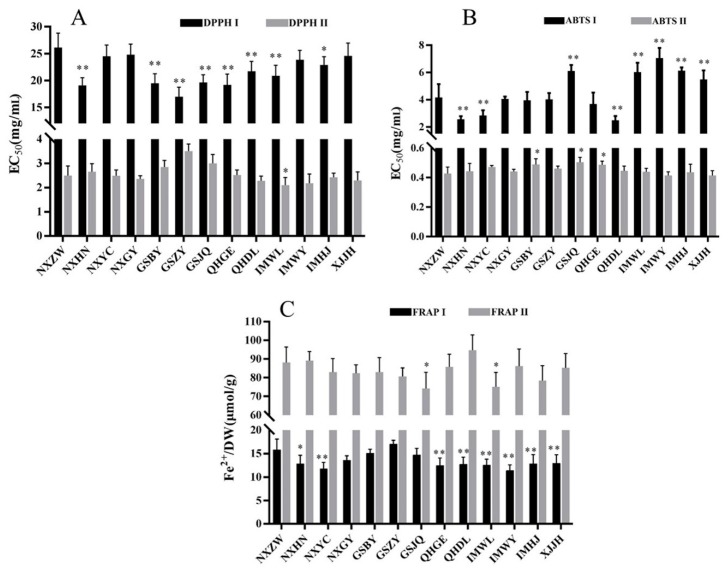
Antioxidant activity for extracts I and II of FLB from different regions ^1,2,3,4^ (^1^ I, hexane/acetone/ethanol (2:1:1, *v*/*v*/*v*) extraction; II, 80% ethanol solution extraction; ^2^ (**A**) EC_50_ of DPPH; (**B**) EC_50_ of ABTS; (**C**) concentrations of Fe^2+^/DW; ^3^ * and ** represent significant differences with the Daodi herb region (NXZW), *p* < 0.05 and *p* < 0.01, respectively; ^4^ NXZW, Zhongwei of Ningxia; NXHN, Huinong of Ningxia; NXYC, Yinchuan of Ningxia; NXGY, Guyuan of Ningxia; GSBY, Baiyin of Gansu; GSZY, Zhangye of Gansu; GSJQ, Jiuquan of Gansu; QHGE, Ge’ermu of Qinghai; QHDL, Delingha of Qinghai; IMWL, Urad Front Banner of Inner Mongolia; IMWY, Wuyuan of Inner Mongolia; IMHJ, Hanggin Rear Banner of Inner Mongolia; XJJH, Jinghe of Xinjiang.

**Figure 3 molecules-24-02228-f003:**
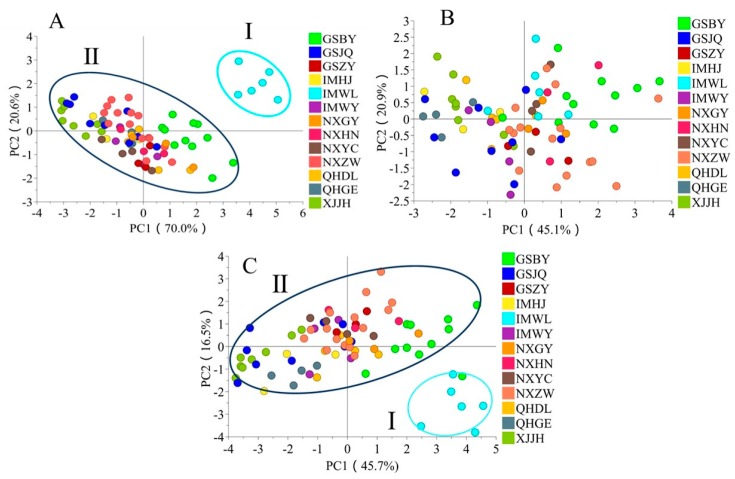
PCA plots of FLB from different regions with functional components ((**A**) PCA plots with total and individual carotenoids; (**B**) PCA plots with the total phenolic, total flavonoid, rutin, ferulic acid, and total polysaccharide; (**C**) PCA plots with total carotenoid and individual carotenoids, total phenolic, total flavonoid, rutin, ferulic acid, and total polysaccharide).

**Figure 4 molecules-24-02228-f004:**
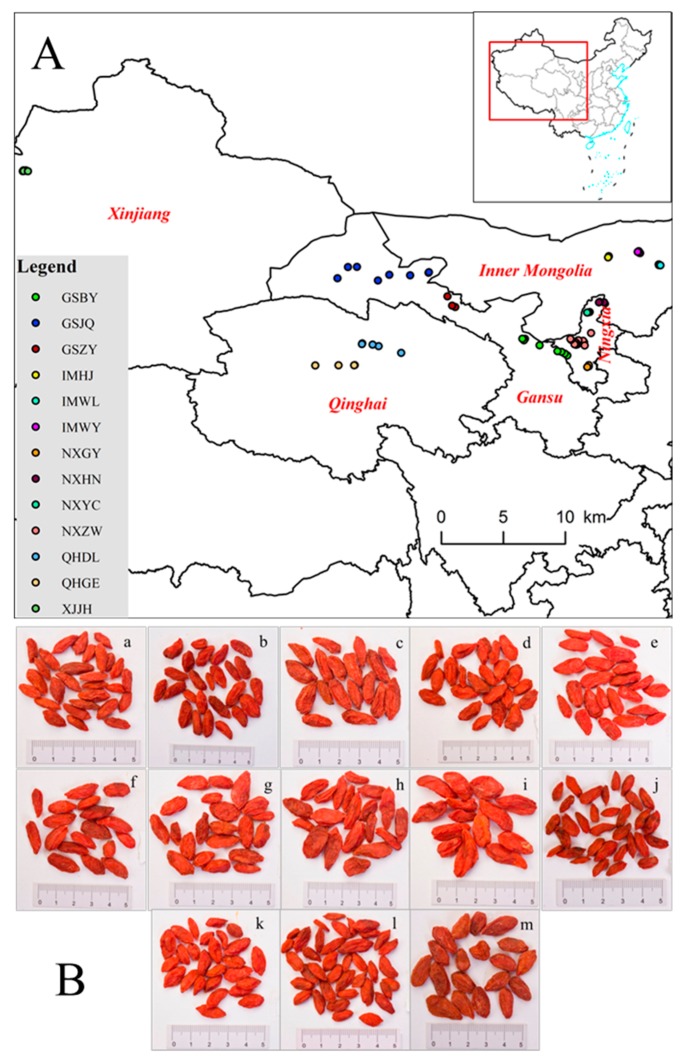
Sample information for the analyzed FLB ((**A**) sample distribution; (**B**) materials of FLB from different regions; a: NXZW; b: NXHN; c: NXYC; d: NXGY; e: GSBY; f: GSZY; g: GSJQ; h: QHGE; i: QHDL; j: IMWL; k: IMWY; l: IMHJ; m: XJJH).

**Table 1 molecules-24-02228-t001:** Functional component contents (mg/g DW) of fruit of *Lycium barbarum* from different regions.

Regions	Contents of Analytes (mean ± SD, *n* ≥ 3)									
	Zeaxanthin	β-Cryptoxanthin	β-Carotene	Zeaxanthin Dipalmitate	TCC ^4^	TFC ^4^	TP ^4^	Rutin	Ferulic Acid	TLBP ^4^
**Daodi herbs region**										
NXZW ^1^	0.0856 ± 0.0216 c ^2^	0.0009 ± 0.0002 cd	0.0615 ± 0.0197 cd	7.375 ± 2.307 abc	17.71 ± 3.94 cde	5.545 ± 0.769 abc	8.253 ± 0.839 a	0.2609 ± 0.0566 cdef	0.1659 ± 0.0405 a	31.41 ± 5.30 bc
**Non-Daodi herbs region**										
NXHN	0.0497 ± 0.0127 de** ^3^	0.0007 ± 0.0002 de	0.0830 ± 0.0376 bcd	8.002 ± 1.024 abc	20.75 ± 3.10 abcd	4.942 ± 0.606 bcd	7.902 ± 0.587 ab	0.2874 ± 0.0517 bcde	0.1453 ± 0.0573 abc	42.45 ± 2.79 a*
NXYC	0.0277 ± 0.0112 e**	0.0003 ± 0.0001 fg**	0.0702 ± 0.0258 cd	8.301 ± 1.491 abc	19.62 ± 2.57 bcd	4.692 ± 0.767 cde*	7.707 ± 0.714 ab	0.3400 ± 0.0654 bc*	0.1305 ± 0.0198 abc	34.19 ± 5.04 b
NXGY	0.0651 ± 0.0164 cd	0.0010 ± 0.0001 cd	0.0980 ± 0.0310 abc	9.766 ± 1.904 ab	21.69 ± 3.87 abc	5.936 ± 0.606 a	7.555 ± 0.301 ab	0.3771 ± 0.0289 ab **	0.1335 ± 0.0132 abc	27.80 ± 2.74 bcd
GSBY	0.1139 ± 0.0157 b**	0.0014 ± 0.0003 b**	0.1133 ± 0.0363 ab**	10.04 ± 2.13 a**	23.71 ± 3.37 ab**	6.144 ± 0.710 a*	7.663 ± 0.477 a*	0.4391 ± 0.0999 a**	0.1726 ± 0.0340 a	31.91 ± 4.51 bc
GSZY	0.0487 ± 0.0192 de**	0.0004 ± 0.0002 f**	0.0788 ± 0.0096 bcd	9.681 ± 1.206 ab	21.25 ± 0.34 abc	5.821 ± 0.303 ab	7.966 ± 0.586 ab	0.2413 ± 0.0309 def	0.1390 ± 0.0328 abc	29.64 ± 1.40 bcd
GSJQ	0.0554 ± 0.0224 de**	0.0006 ± 0.0002 ef**	0.0549 ± 0.0263 de	5.565 ± 2.531 cd	14.72 ± 4.05 f	4.737 ± 0.610 cde*	7.443 ± 0.907 ab*	0.1812 ± 0.0727 f**	0.1147 ± 0.0289 bc**	30.161 ± 5.65 bcd
QHDL	0.0324 ± 0.0171 e**	0.0010 ± 0.0001 c	0.0544 ± 0.0101 de	7.757 ± 1.814 abc	17.09 ± 2.06 cde	3.859 ± 0.540 ef*	7.711 ± 0.563 ab	0.3283 ± 0.0763 bcd*	0.1122 ± 0.0225 bc*	27.54 ± 5.25 bcd
QHGE	0.0317 ± 0.0193 e**	0.0011 ± 0.0001 c*	0.0752 ± 0.0208 cd	9.403 ± 2.732 ab	20.73 ± 3.28 abcd	3.546 ± 0.362 f*	6.960 ± 0.237 b*	0.211 ± 0.0559 ef	0.1027 ± 0.0184 c**	23.62 ± 2.15 d*
IMWL	0.2857 ± 0.0190 a**	0.0031 ± 0.0002 a**	0.1301 ± 0.0307 a**	10.07 ± 1.65 a*	25.35 ± 1.91 a**	4.906 ± 0.455 bcd*	7.176 ± 0.624 ab*	0.3457 ± 0.0479 bc**	0.1588 ± 0.0300 ab	30.35 ± 4.81 bcd
IMWY	0.0538 ± 0.0105 de**	0.0005 ± 0.0001 ef**	0.0672 ± 0.0204 cd	8.300 ± 2.001 abc	20.12 ± 3.47 bcd	5.977 ± 0.646 a	7.219 ± 0.860 ab*	0.2357 ± 0.609 def	0.0994 ± 0.0180 c**	25.13 ± 2.50 cd*
IMHJ	0.0482 ± 0.0243 de**	0.0005 ± 0.0001 ef**	0.0607 ± 0.0211 cd	6.654 ± 2.078 bcd	15.78 ± 2.40 ef	4.011 ± 0.598 def*	6.899 ± 0.765 b*	0.2566 ± 0.0093 cdef	0.1059 ± 0.0228 c*	25.17 ± 3.64 cd*
XJJH	0.0429 ± 0.0252 de**	0.0001 ± 0.0000 g**	0.0229 ± 0.0090 e**	4.260 ± 1.451 d**	12.93 ± 2.87 f**	3.177 ± 0.659 f*	7.239 ± 0.803 ab*	0.2274 ± 0.0343 ef	0.1309 ± 0.0269 abc*	26.78 ± 3.88 cd*

^1^ NXZW, Zhongwei of Ningxia; NXHN, Huinong of Ningxia; NXYC, Yinchuan of Ningxia; NXGY, Guyuan of Ningxia; GSBY, Baiyin of Gansu; GSZY, Zhangye of Gansu; GSJQ, Jiuquan of Gansu; QHGE, Ge’ermu of Qinghai; QHDL, Delingha of Qinghai; IMWL, Urad Front Banner of Inner Mongolia; IMWY, Wuyuan of Inner Mongolia; IMHJ, Hanggin Rear Banner of Inner Mongolia; XJJH, Jinghe of Xinjiang. ^2^ Values followed by the same letter in the same column are not significantly different (*p* > 0.05).^3^ * and ** represent significant differences with Daodi-herb region (NXZW), *p* < 0.05 and *p* < 0.01, respectively. ^4^ TCC, TFC, and TPC contents were expressed as mg βCE/g DW, mg RE/g DW, and mg GAE/g DW.

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
