# Peer review of "Comparison of Functional Components and Antioxidant Activity of Lycium barbarum L. Fruits from Different Regions in China"

_molecules, 2019, doi:10.3390/molecules24122228_

Round 1
Reviewer 1 Report
After careful evaluation of this manuscript, I have the following suggestions for the authors to improve their manuscript.
In overall, the authors should beconsistent through the manuscript, use the same terminology and abbreviations. They should also not introduce new terminology in different parts of manuscript (for instance extract I and extract II introduced at the end in results of antioxidant activity) and be clear on each section, units, and whether the results obtained from a spectrophotometric or HPLC analysis. This is especially important for carotenoid section as both spectrophotometric and HPLC methods used.
Please revise to“Lycium barbarum L.” when first mentioned in the text and in the title.
There is still need for proof-reading. Please go over whole manuscript for its consistency and language.
An abbreviation section should be added.
Abstract/Introduction
Line 24: “All of these FLBs could be divided into two clusters in PCA analysis, and individual carotenoids/total carotenoid contents could be used as the markers for the regional characterization.
Line 28: “…Daodi-herbs which are commercially available standardized FLB.” This sentence requires some more explanation.
Line 30: Conventional name of the fruits may be added (gouji or wolfberries etc.) in Keyword sections to increase the appearance of the manuscript in web searches. Since Latin name (L. barbarum), has already been given in title and abstract, it can be omitted from the list.
Line 50: “In recent years, the demand for FLB is increasing due to its potential beneficial effects on human “. If available, adding some market/export values might be helpful to show its potential.
Results and Discussion:
Line 69-80; 317: These part presenting optimization data for carotenoid extraction is satisfactory enough. However, the presented ratios are not solvents volumes (Line 77), instead they are sample:solvent ratios. Furthermore, the final optimized conditions should be given using the same terminology. “….0.5 g fruit powder were mixed with hexane/acetone/ethanol in 1:80 sample:solvent ratio…..”
Line 86: “… better separationthan….” Or “the best separation…’ better require a comparison
Line 88: “validated for its….”
Line 94: Is there any reference/guideline that the authors used to compare their validation results to conclude their results are satisfactorily repeatable and accurate?
Line 98: Results for total carotenoid content and individual carotenoids should be presented and discussed separately and in order in this section. Otherwise, it is very confusing and hard to follow. Total carotenoid content by spectrophotometer should be evaluated apart from individual carotenoids. The authors can sum up individual carotenoids and discuss total carotenoid content by HPLC, separately, if they are willing to do so. Total carotenoid contents were not compared with the literature values, and therefore, they were not discussed properly. How could the authors explain the big difference between the sum of individual carotenoids and TCC?
Line 134: There is no place for speculation in a scientific paper. “Owing to the characteristic of carotenoids, it might be attributedspeculatedthat neoxanthin in saponified extract might be degraded to/transformedfromto other components (which possible components? Epoxides?) during the process of saponification “. Please explain what is expected from this sentence. Is it complete degraded?
Line 137: “It was recognized that the curative effect of Daodi-herbs is better than non Daodi-herbs” The authors did not test curative effects, they have just tested carotenoids. So, this sentence is not correct. If curative effect of these herbs is attributed to their carotenoid contents, then it should be written like this a reference should be cited showing their relation.
Line 141: “….significantly different …“ higher or lower?
Line 143: Is (P>0.05) means different or no difference?
Line 146: It is “total phenolic content”, not “phenolic acids”
Line 150: “To find whether the color of the extract affect the determination results of phenolic, two extraction methods were compared “. A new methodology was given here although it is not given in methods section. It should be mentioned in methods section as well.
Lien 165: Total phenolic and flavonoid contents should be compared with literature values and discussed if the results similar to or higher/lower than literature. The same applies for other sections as well.
Line 263: “The relationship between carotenoids content and antioxidant activity was measured based on Pearson’s Correlation Coefficient Test.“ Did the correlations established just with carotenoid contents or also with phenolic and flavonoid contents? Also, correlation is not a measurement. Different extracts were used for carotenoid and phenolic analysis, as well as antioxidant activities (Extract I and Extract II). Which results are correlated here in Table 2? Correlations should be established with ethanol extracts versus phenolic/flavonoid contents and other organic extract versus carotenoids. Please revise the table and text thoroughly to explain these differences clearly. It is also still questionable to establish correlation between EC50 values and total contents of somethings as the higher the phenolic content the lower the EC50 values.
Figure 1: Which sample does the chromatogram belong to?
Figure S1 and S2: Standard deviation bars should be added to the figures. What are the number of replicates? It should also be mentioned in methods section that optimization of carotenoid extraction were evaluated based on individual carotenoids (not TCC).
Figure S3. The authors are advised to add representative chemical structures for phenolic compounds (gallic acid/rutin).
Table S1: Please use the same number of significant units while reporting your results.
Table S3 and S4: Abbreviations used in tables should be explained at least as footnote. Because tables should be stand-alone without text. If the authors did not detect neoxanthin and lutein in none of their samples, they are suggested to exclude them from the table for simplification and may be keep this information just in the text part.
Methods:
HPLC methodology for the analysis of individual phenolic compounds should be given either in methods section or supplementary data.
Line 298: Extrasynthedse
Line 301: “… series of diluted standardssolutionswere prepared in thefollowing concentration ranges for HPLC-DAD analysis:
Line 310: “In UV analysis,For spectrophotometric analysis of carotenoid contents, a series of β-carotene were prepared in the concentration range 2.13-42.47 μg/ml. “. UV analysis does not necessarily represent spectrophotometric analysis. DAD is also a kind of UV.
Line 303: “Determination oftotal carotenoid contents and individual carotenoids”
Lien 304: “Preparation of standard solutions”. The same applies for other similar cases.
Line 312: Since carotenoids are susceptible to degradation during storage (especially 4 C), it might be beneficial to write maximum times of storage.
Line 314: The subheading should be updated as “Carotenoid extraction” or “Extract preparation for the analysis of carotenoids” rather than “Sample solution preparation”
Line 316: “According tosingle factor experiment design….” Which sample were used for optimization studies? NXZW as in HPLC method section?
Line 322: Which material of membrane filter were used? The same applies for other similar cases.
Line 324-326: Separation between total carotenoid content determined by spectrophotometer and HPLC should be done clearly both in abstract/results and method sections. Here are some suggestions this part:
The titles might be updated accordingly:
“Analysis of total carotenoid contents (TCC) by spectrophotometry” (Line 324)
“Analysis of individual carotenoids by HPLC-PDA” (Line 328).
“Methodvalidation for the analysis of individual carotenoids” (Line 336)
PDA and DAD are used interchangeably throughout the manuscript. Please use one of them consistently.
Method validation include both extraction and HPLC.
It is advisable to use open form for abbreviations in headings (for instance; TCC for total carotenoid contents). To separate carotenoid contents determined by spectrophotometer and HPLC, the authors may express spectrophotometric-based total carotenoid contents as “mg/g beta-carotene eq. (BCE) DW”in every case throughout the manuscript (Line 324-327). The units for all other spectorophotometric assay should also be updated accordingly, for instance, mg/g GAE DW for total phenolics and mg/g rutin equivalents (RE) DW for total flavonoids. Otherwise, it is hard to understand whether the results are from a HPLC or spectrophotometric analyses (Lines 369-382)
Line 340-343: What is the difference between repeatability and precision? How did the authors calculate repeatability as it is the same thing with intra/inter-day variability? Did they use the same sample extracts for precision? If yes, then precision is related with HPLC analysis and it is “HPLC precision” Please explain it clearly. Linearity is also a part of validation as mentioned in Line 88
Line 244: Stability part should start in a new paragraph
Line 353: “Determination(or Analysis)of total phenolic and flavonoids contents” It is total phenolic content, not total phenolic acid. Please be consistent in language with the previous carotenoid section
Line 355: “Stock solutions of the rutin and gallic acid standards were prepared in 80% ethanol at a concentration of 2 mg/ml. In UV analysis,A series of rutin were prepared in the concentration range 0.2880-0.8640 mg/ml, and gallic acid concentration at a range of 0.2829-0.9902 mg/ml. “ Please use suitable significant figures for numbers
Line 350: Since this section is almost the same with carotenoid extraction except solvents used, it is better to combine them in one subsection (3.4) to avoid repetition and introduce terminology of Extract I and Extract II in here. The same applies for the preparation of standard solutions (as section 3.3)and extract preparation for polysaccharides (if possible).
Line 371: “An aliquot of 20 ul of respective standards or 80% ethanol extracts (Extract II) were mixed with ….”It is always better to point out clearly which extract were used in which section. If exactly the same Folin method used with the reference, then there is no need to explain procedure. Only the modifications should be given. The same applies for other assay as well.
Line 397: What is TLBP? Please use long forms and abbreviations for headings. Also be consistent with the language (Determination, analysis or measurement) through the manuscript
Line 399: “…0.5 mL of glucose standard..” Since there are many standards and methods, it is better to be very clear.
Line 406: Please add more information here about which extracts used for antioxidant capacity assays (extract 1 and extract II). Terminology for Extract I and Extract II should be established in methods section (subsection 3.4 as mentioned above) first. Please explain all terminology in order first in methods section and do not introduce new terminology later. For instance, we see first see Extract 1 and Extract II in results section of antioxidant activity. In addition, there is no need to explain the detailed procedures if exactly the same procedure applied with the references. Please only give modification, and how the results expressed.
Could the authors explain what does EC50 (mg/ml DW) mean? Is it mg extract or mg samples? What does mL DW stand for? Dry weight mL extract?
Conclusion
Line 432: functional componentscontents
Line 436: total carotenoid contents or individual carotenoids?
Regarding responses to reviewer 2:
Comment 1.The authors say that they did not present their HPLC results for phenolic compounds because they are degraded during drying process and their level is very low (70 ug/g). However, the levels of some individual carotenoids are in the similar level (for instance; zeaxanthin:0.0856 mg/g DW for NXZW=85 ug/g DW). Thus, elimination of these data seems rather arbitrary. I would suggest to the authors including HPLC analysis for phenolic compounds in the manuscript (both in method and results sections), and discuss it properly why it is very low and should not be included into account in correlation/PCA etc. They are strongly advised to include this discussion to the manuscript that they have done for responding to the reviewer.
Comment 4 and 5.I clearly agree with the authors and know that the results can be expressed as EC50 values or mol/g DW based on the literature. However, it is also possible to find antioxidant activity data in other units and the authors should be expressed their results in a meaningful way for their manuscript and for their discussion. There is no sense to seek correlation between EC50 and mol/g DW in antioxidant assays and mg/g DW values for other spectrophotometric assay. The higher the phenolic content is the lower the EC50 values, i.e. the efficient concentration (of an extract which will contain higher amounts of phenolic compounds when TPC is higher) to inhibit 50% of DPPH radical. If the authors wanted to present something else that I have understood, they should make it clear in the manuscript. Similarly, mol (as expressed in FRAP) and gram (as expressed in TPC) of a specific molecule/a mixture of molecules, are clearly different things. Therefore, either antioxidant activity and other results should be evaluated separately, or the authors should try to express their results in a comparable way.
Author Response
Dear reviewer,
Thank you for your constructive and helpful comments and suggestions on the manuscript (molecules-508068) provided by us for publication on Molecules. We have studied your comments carefully and have tried our best to revise and improve our manuscript according to the comments. We hope you will find the improved manuscript worthy of consideration for publication in Molecules. The detailed revisions and corrections are listed in the Word document point by point.

Reviewer 2 Report
Manuscript Molecules-508068 requires some changes to improve its scientific quality:
Folin-Ciocalteau method is widely used to measure the total content of plant phenolics (flavonoid and non-flavonoid phenolics) in plant extracts. Authors suppose that have measure only phenolic acids, but that is not true. Please, use the term “Total phenolics”, not “Total Phenolic Acids” for results obtained by Folin-Ciocalteau method throughout the text. Moreover, change the caption of subsection 3.4 as follows: “Determination of Total Phenolics (TPC) and Total Flavonoid Contents (TFC)”, and explain why the correlations observed between TPC and antioxidant activities are not observed for TFC (subsection 2.6 and table 2), giving some hypothesis to explain these differences.
Authors use “the significant difference”. This is incorrect. Use “significant difference” or “significant differences” (e.g., lines 143, 227)
Please, rewrite sentence in lines 205-206.
What means TLBP? (line 296).
Author Response

(The authors gave the same response as above.)

Round 2
Reviewer 1 Report
The scientific quality of the manuscript is appropriate for publication after minor improvements as listed below. I kindly as to the authors to carefully review all manuscript for each comment, not for he mentioned lines, both also for the all similar cases.
1) HPLC data for phenolic compounds is in important part to be mention:As understood, the authors do not want to include any data about HPLC analysis and individual phenolics. If they do not have a whole data set for HPLC analysis of phenolic compounds, they can give the values for the ones they have measured, and can say that ’they preferred to focus on XXX assays” after judging the necessity of this approach. However, now, authors judgements do not make any sense. They claim that individual phenolic compounds are degraded during drying that they could not detect them in HPLC. However, later they suggest that phenolic compounds as determined by TPC are responsible from antioxidant activity.
Line 221: Please no speculation! Provide a reference for this assumption showing that drying reduces concentration of phenolic compounds to undetectable levels. Please also note that detection limits for HPLC are generally on ng levels. These two phenolics are above the limit of detection. Please give values for the detected two phenolics.
2) One of the main language errors is for instance “the contents of carotenoids” although it is introduced as “total carotenoid contents (TCC)” in other places. The same applies for all other similar cases. Another issue is unnecessary plural words.
Other suggestions are as listed below (mostly including correction on expression and language corrections):
Line 19: “In this study, the contents ofcarotenoids, phenolic, flavonoid, and polysaccharidescontents…….” Please revise in other cases in all manuscript (Line 22 etc.)
Line 20: “….from 13 different regions in China in total of 78samples….”
Line 21: “total carotenoidscontent” carotenoid should be singular. Please delete ‘s’ The same applies for total phenolic, polysaccharides, flavonoid contents. Please correct through the manuscprit.
Line 21: “total carotenoidscontents were from 12.93 to 25.35 mg/g BCB DW”. Please check all units and revise whenever necessary.
Line 21-22: “Zeaxanthin dipalmitate was the predominant carotenoid (XX mg/g DW) in FLB”. Please always refer to the numbers.
Line 25: “…. bothindividual carotenoids and total carotenoids contents ….”
Line 28: “Considering the functional components contentsand antioxidant activities”
Line 29: “…which are commercially available high qualityFLB.” Are they standardized or high quality?
Line 29: “The results of this study could provide….”
Line 22-42: Abbrevations: extract I and extract II are not a part of abbrevations
Line 39 and other similar cases: Is it βCE, β-carotene equivalents; or BCB? Please check all units for other assays as well. For instance, for phenolic contents, if the unit is mg GAE/g DW, always use this unit. Not mg/g DW.
Line 44: “goji or wolfberry“ Both should be singular.
Line 46: “In traditional Chinese medicine, it is believed that FLB could nourish liver and kidney, and improve the eyesight “
Line 52: “total of 11 carotenoids such as ……(please give examples) and 53 polyphenols such as……(please give examples)Theprevious studies of bioactive components in FLB showed that carotenoids from FLB….could enhance functioning of immune system and prevent the chronic diseases, and possess biological activities such as antiaging, antidiabetic, anticancer, and so on [10-13]. Phenolic compoundsfrom FLB showed the properties ofantioxidant, antiaging, antiinflammatory, and antihyperglycemic properties [14-16]. Furthermore, FLP polysaccharides exhibited a variety of functions including antiaging, antitumour, antidiabetic, immunomodulation, hypoglycaemic and hypolipidaemic activities [12, 17-19]. Thus, carotenoids, phenolic compounds and polysaccharides in FLB have received increasing attention for theirhealth promoting properties [11, 20, 21]. “m It is important out point out whether these previous studies conducted on FLB. Be consistent on using dash in anti ces: for instance: antitumour or anit-tumour.
Line 61-62 please give a reference for this sentence including export values.
Line 65: “Daodi-herbs……. are widely recognized as the symbol of good quality with better beneficial clinical effect than those of non Daodi-herbs produced in other regions.”
Line 69-72: “…there are few studiesabout whether contentd of other functional constituents such as XXXX (Please fill) ……and the antioxidantactivities such as the antioxidationof FLBs produced in different regions are different, and which regions have potential to produce high quality FLBs with similar functional composition to those of high quality Daodi-herbs. “Are there few studies? Which of them"? Or “there is a lack of studies on….”
Line 73-76: “In the present study, carotenoid, flavonoids, phenolic and polysaccharide contents and antioxidant activities of FLBs produced in different regions were determined and compared. The correlation analysis among contents and antioxidant activities was assessed.“ Do you still have correlations?
Line 80: Sample solution preparation: Could the authors justify the difference between a solution and an extract? Extract preparation is more appropriate term for these sections. Although they later label their “sample solutions” as “extract”, the subheadings (2.1.1, 2.2.1, ) are still “sample solution preparation”. Please be consistent through the all manuscript.
Line 80: “…extracted … to extract…” Please revise this sentence
Line 86: What is “maximum extract percentages”? It should be “maximum extraction”
Line 91: “…was optimumconditions for extraction of individual carotenoids
Line 94: It is “reversed phase chromatography with C30 column”. Not “reversed phase columns chromatography”.
Line 102: All analytes calibration curves
Line 114: It is “total carotenoid content (TCC)” as introduced in method section. Not “total contents of carotenoids” Do you have different content values for carotenoids for TCC?
Line 131:allthe highest
Line 136-139: “In addition, it was notable hat the significant difference existed between the sum of individual carotenoids (4.326-10.49 mg/g DW) and TCC (12.93-23.35 mg/g DW), which suggested that some carotenoids exist in FLB besides these six compounds analyzed in this assay” What do you means in this sentence, You method is not good enough to detect all carotenoids although the method is optimized? Could you please provide a reference for this assumption? Is there any methodological limitation, interferences in spectrophotometer?
Line 176: Why is the faint yellow color is important? Is it because it might be related with carotenoids or phenolics?
Line 177: optimizedmethod
Line 183: “Thus, ethanol solution was used to extract phenolic compounds directly” Both extractions provide ethanol solutions to provide phenolic compounds directly. So, which one selected?
Line 202: “was consistent with the reported values by…”“new goji cultivars” What does new goji cultivar mean? New cultivar produce din XX or new cultivar rich in XX?
Line 250 : “Antioxidant Activities of FLBs from Different Regions” Plural
Line 251-260: Please clarity what is mg/mL in EC50 in method section. Is it mg DPPH/mL extract or mg extract/ mL DPPH or something else?
Line 250: please provide comparison of antioxidant activities by literature values as much as possible
Line 306: ” at least 3sets of samples were collected in the same regions” Is this 3 set different than three biological replicates? Is it 3 biological replicate x 3 some other replicates?
Line 308: until theconstant weight not change
Line 309-310: Why were the authors analyzed moisture content and for which samples (fresh or dried until constant weight)? Please specify. In addition, moisture values should be reported individually for all samples or as a range in somewhere in the manuscript.
Line 312: methyl tert-butyl ether (MTBE)
Line 318: EXTRASYNTHASE . Please always confirm what is suggested to you.
Line 320: How did they evaluated HPLC purities? Based on the peak areas, and then subtracting the area of impurities, or these impurities are specified by supplier?
Line 329: “certain amounts of individual compounds (similar for Line 383)”
Line 335: “a series dilutionsof β-carotenesolutions”
Lien 337: membrane material is important here, PTFE, PVDF, CE, etc.
Line 371: what is sample 5? If it is the sample number from a Table, please refer to the table.
Line 382: This section is too long and confusing. It is advised to separate sections as extract preparation, optimization and analysis as done in carotenoid section.
Line 371-392: Please rewrite this paragraph (6 lines for one sentence) with short and clear sentences. Please do not use unnecessary parenthesis. Which method were used for further optimization and optimized extraction?
Line 406: “Analysis of Polysaccharide contents.” Plural since other headings are also plural and you have many samples.
Line 424: Antioxidant activity or capacity? Please be consistent. “Antioxidant activity assays.” (Plural). Please specify which extracts were used for antioxidant activity assays here. “Extract I and Extract II as prepared in section X and X, respectively, were used for all antioxidant activity assays”
Line 425: “Brand-Williams“ Please check the manuscript for typing errors.
Author Response
Response to Reviewer #1:
1. HPLC data for phenolic compounds is in important part to be mention:As understood, the authors do not want to include any data about HPLC analysis and individual phenolics. If they do not have a whole data set for HPLC analysis of phenolic compounds, they can give the values for the ones they have measured, and can say that ’they preferred to focus on XXX assays” after judging the necessity of this approach. However, now, authors judgements do not make any sense. They claim that individual phenolic compounds are degraded during drying that they could not detect them in HPLC. However, later they suggest that phenolic compounds as determined by TPC are responsible from antioxidant activity.
Line 221: Please no speculation! Provide a reference for this assumption showing that drying reduces concentration of phenolic compounds to undetectable levels. Please also note that detection limits for HPLC are generally on ng levels. These two phenolics are above the limit of detection. Please give values for the detected two phenolics.
Reply: Thanks for your comments. Done, the results of rutin and ferulic acid contents have provided, please see Table 1 and Table S5.
In this study, rutin contents were varied from 0.1812 to 0.4391 mg/g DW, and ferulic acid contents were from 0.0994 to 0.1726 mg/g DW. Rutin content in samples from GSBY was the highest, followed by the samples from NXGY. Ferulic acid content in FLB from different regions, from high to low was: GSBY > NXZW > IMWL > NXHN > GSZY > NXGY > XJJH > NXYC > GSJQ > QHDL > IMHJ > QHGE. Zhang et al. reported that rutin and ferulic acid contents in FLB were 67.9 μg/g FW and 152.2 μg/g FW, respectively, which were much higher than the results in this study, taken the moisture content in fresh samples of FLB as 70%-75%. In addition, the results in this study were also consistent with the reference reported that individual phenolic acid and flavonoid contents in the dried samples of FLB were as low as microgram level. Please see lines 223-233.
2. One of the main language errors is for instance “the contents of carotenoids” although it is introduced as “total carotenoid contents (TCC)” in other places. The same applies for all other similar cases. Another issue is unnecessary plural words.
Reply: Thanks for your comments. Done, please see lines 19, 23, 151-162, 295 and 386.
3. Other suggestions are as listed below (mostly including correction on expression and language corrections):
Line 19: “In this study, carotenoid, phenolic, flavonoid, and polysaccharide contents…….” Please revise in other cases in all manuscript (Line 22 etc.)
Reply: Thanks for your suggestions. Done, please see lines 19, 21, 23, 27, 40, 161-162, 295 and 386.
4. Line 20: “….from 13 different regions in China in total of 78 samples….”
Reply: Thanks for your suggestion. Done, please see line 20.
5. Line 21: “total carotenoid content” carotenoid should be singular. Please delete ‘s’ The same applies for total phenolic, polysaccharides, flavonoid contents. Please correct through the manuscprit.
Reply: Thanks for your suggestions. Done, please see lines 21, 27, 40, 128, 161, 305-307, 336, 393 and 457.
6. Line 21: “total carotenoid contents were from 12.93 to 25.35 mg/g BCB DW”. Please check all units and revise whenever necessary.
Reply: Thanks for your suggestion. Done, please see lines 22-24 and 140.
7. Line 21-22: “Zeaxanthin dipalmitate was the predominant carotenoid (XX mg/g DW) in FLB”. Please always refer to the numbers.
Reply: Thanks for your comment. Done, Zeaxanthin dipalmitate was the predominant carotenoid (4.260-10.07 mg/g DW) in FLB. Please see lines 22-23.
8. Line 25: “…. Both individual carotenoids and total carotenoids contents ….”
Reply: Thanks for your suggestion. Done, please see line 27.
9. Line 28: “Considering the functional components - and antioxidant activities”
Reply: Thanks for your suggestion. Done, please see line 29.
10. Line 29: “…which are commercially available high quality FLB.” Are they standardized or high quality?
Reply: Thanks for your comment. Done, it is generally considered that the quality of Daodi-herbs are high. Please see line 31.
11. Line 29: “The results of this study could provide….”.
Reply: Thanks for your suggestion. Done, please see line 31.
12. Line 22-42: Abbrevations: extract I and extract II are not a part of abbrevations.
Reply: Thanks for your suggestion. Extract I and extract II have been removed from the section of abbrevations.
13. Line 39 and other similar cases: Is it βCE, β-carotene equivalents; or BCB? Please check all units for other assays as well. For instance, for phenolic contents, if the unit is mg GAE/g DW, always use this unit. Not mg/g DW.
Reply: Thanks for your comments. It is βCE, β-carotene equivalents. Please see lines 22, 24 and 363.
14. Line 44: “goji or wolfberry“ Both should be singular.
Reply: Thanks for your suggestion. Done, please see line 45.
15. Line 46: “In traditional Chinese medicine, it is believed that FLB could nourish liver and kidney, and improve the eyesight “
Reply: Thanks for your suggestion. Done, please see lines 47-48.
16. Line 52: “total of 11 carotenoids such as ……(please give examples) and 53 polyphenols such as……(please give examples)The previous studies showed that carotenoids from FLB….could enhance functioning of immune system and prevent the chronic diseases, and possess biological activities such as antiaging, antidiabetic, anticancer, and so on [10-13]. Phenolic compounds from FLB showed the antioxidant, antiaging, antiinflammatory, and antihyperglycemic properties [14-16]. Furthermore, FLP polysaccharides exhibited a variety of functions including antiaging, antitumour, antidiabetic, immunomodulation, hypoglycaemic and hypolipidaemic activities [12, 17-19]. Thus, carotenoids, phenolic compounds and polysaccharides in FLB have received increasing attention for their health promoting properties [11, 20, 21]. “m It is important out point out whether these previous studies conducted on FLB. Be consistent on using dash in anti ces: for instance: antitumour or anit-tumour.
Reply: Thanks for your suggestion and comments. As your suggestions, the previous studies conducted on FLB have been point out, and it has been consistent on using dash in anti. Please see lines 51-62.
17. Line 61-62 please give a reference for this sentence including export values.
Reply: Thanks for your comment. Done, the reference has been given, please see lines 65 and 524-525.
18. Line 65: “Daodi-herbs……. are widely recognized as the symbol of good quality with better beneficial clinical effect than those of non Daodi-herbs produced in other regions.”
Reply: Thanks for your suggestion. Done, please see line 69.
19. Line 69-72: “…there are few studies about whether contentd of other functional constituents such as XXXX (Please fill) ……and the antioxidant activities-of FLBs produced in different regions are different, and which regions have potential to produce high quality FLBs with similar functional composition to those of high quality Daodi-herbs. “Are there few studies? Which of them"? Or “there is a lack of studies on….”
Reply: Thanks for your suggestion and comments. Done, the sentence has been revised as: “there is a lack of studies on : whether the profile of the other functional constituents such as carotenoids, phenolic, flavonoid et al. and the antioxidant activities of FLBs produced in different regions are different; and which regions have potential to produce high quality FLBs with similar functional composition to those of high quality Daodi-herbs”. Please see lines 72-76.
20. Line 73-76: “In the present study, carotenoid, flavonoids, phenolic and polysaccharide contents and antioxidant activities of FLBs produced in different regions were determined and compared. The correlation analysis among contents and antioxidant activities was assessed.“ Do you still have correlations?
Reply: Thanks for your comment. It is our mistake, the correlation analysis has been removed, and this sentence has been deleted.
21. Line 80: Sample solution preparation: Could the authors justify the difference between a solution and an extract? Extract preparation is more appropriate term for these sections. Although they later label their “sample solutions” as “extract”, the subheadings (2.1.1, 2.2.1, ) are still “sample solution preparation”. Please be consistent through the all manuscript.
Reply: Thanks for your suggestion. Done, it is corrected as extract preparation, please see lines 83 and177.
22. Line 80: “…extracted … to extract…” Please revise this sentence
Reply: Thanks for your suggestion. Done, the sentence has been revised as “Carotenoids are conventionally obtained with non-polar solvents (e.g., hexane, petroleum ether and tetrahydrofuran) for extracting non-polar carotenes or esterified xanthophyll, and polar solvents (e.g., acetone, ethanol, ethyl acetate) for extracting polar carotenoids.” Please see lines 84-86.
23. Line 86: What is “maximum extract percentages”? It should be “maximum extraction”
Reply: Thanks for your suggestion. Done, it has been revised as “maximum extraction”, please see line 89.
24. Line 91: “…was optimum conditions for extraction of individual carotenoids.
Reply: Thanks for your suggestions. Done, please see line 94.
25. Line 94: It is “reversed phase chromatography with C30 column”. Not “reversed phase columns chromatography”.
Reply: Thanks for your suggestion. The sentence is revised as “It has been reported that reversed phase chromatography with C30 column gets a better separation for carotenoids.” Please see lines 96-97.
26. Line 102: All calibration curves
Reply: Thanks for your suggestion. Done, please see lines 103-104.
27. Line 114: It is “total carotenoid content (TCC)” as introduced in method section. Not “total contents of carotenoids” Do you have different content values for carotenoids for TCC?
Reply: Thanks for your suggestion. Done, it is revised as “total carotenoid content (TCC)”, please see line 116.
28. Line 131: the highest
Reply: Thanks for your suggestion. Done, please see line 133.
29. Line 136-139: “In addition, it was notable that the significant difference existed between the sum of individual carotenoids (4.326-10.49 mg/g DW) and TCC (12.93-23.35 mg/g DW), which suggested that some carotenoids exist in FLB besides these six compounds analyzed in this assay” What do you means in this sentence, You method is not good enough to detect all carotenoids although the method is optimized? Could you please provide a reference for this assumption? Is there any methodological limitation, interferences in spectrophotometer?
Reply: Thanks for your comments. Inbaraj et al. (Journal of Pharmaceutical and Biomedical Analysis: 10.1016/j.jpba.2008.04.001) identified 11 free carotenoids (including five zeaxanthin isomers, four β-carotene isomers, neoxanthin and β-cryptoxanthin) and seven carotenoids esters (including three zeaxanthin monopalmitate isomers, three β-cryptoxanthin monopalmitate isomers and one zeaxanthin dipalmitate) from FLB extract by HPLC-DAD-APCI-MS. Thus, although the method is optimized, it was not good enough to detect all carotenoids due to the disadvantages of HPLC-DAD.
This assumption is provided base on Inbaraj et al. (Journal of Pharmaceutical and Biomedical Analysis: 10.1016/j.jpba.2008.04.001) reported. It has been added, please see lines 140-141.
UV spectrophotometry has poor selectivity in quantitative analysis, due to the presence of other chemical compounds that absorbed electromagnetic radiation (Food Chemistry: 10.1016/j.foodchem.2018.08.060). It has been added, please see lines 142-144.
30. Line 176: Why is the faint yellow color is important? Is it because it might be related with carotenoids or phenolics?
Reply: Thanks for your comments. Different concentrations of FLB carotenoids solutions appear yellow to red color. Thus, the faint yellow color might be related with carotenoids. Morover, the determination of total phenolic content by spectrophotometry is based on color reaction. Thus, it is necessary to define whether the faint yellow affects the chromogenic reaction of total phenolic determination.
31. Line 177: optimized method
Reply: Thanks for your suggestion. Done, please see line 181.
32. Line 183: “Thus, ethanol solution was used to extract phenolic compounds directly” Both extractions provide ethanol solutions to provide phenolic compounds directly. So, which one selected?
Reply: Thanks for your comment. The sentence has been revised as “Thus, 80% ethanol solutions was used to extract phenolic compounds with ultrasonic bath directly.” Please see lines 187-188.
33. Line 202: “was consistent with the reported values by…“new goji cultivars” What does new goji cultivar mean? New cultivar produced in XX or new cultivar rich in XX?
Reply: Thanks for your comment. This sentence has been revised as “TPC in FLB was consistency with the report by Zhao et al., and was higher than new goji cultivars which produced in Poland.” Please see lines 206-207.
34. Line 250 : “Antioxidant Activities of FLBs from Different Regions” Plural
Reply: Thanks for your suggestion. Done, please see line 260.
35. Line 251-260: Please clarity what is mg/mL in EC50 in method section. Is it mg DPPH/mL extract or mg extract/ mL DPPH or something else?
Reply: Thanks for your comment. Done, it is the mg sample/ml, please see lines 265-267.
36. Line 250: please provide comparison of antioxidant activities by literature values as much as possible
Reply: Thanks for your comment. In this study, the results of antioxidant capacity were expressed as EC50 (mg sample/ml). The antioxidant activity of FLB reported in literature was expressed as mg extract/ml. It is difficult to make a comparison between this two expressions.
37. Line 306: ” at least 3 sets of samples were collected in the same regions” Is this 3 set different than three biological replicates? Is it 3 biological replicate x 3 some other replicates?
Reply: Thanks for your suggestion and comment. Done, please see line 316.
38. Line 308: until constant weight
Reply: Thanks for your suggestion. Done, please see line 318.
39. Line 309-310: Why were the authors analyzed moisture content and for which samples (fresh or dried until constant weight)? Please specify. In addition, moisture values should be reported individually for all samples or as a range in somewhere in the manuscript.
Reply: Thanks for your comments. Analyzed moisture content aimed at caculate the dry weight of the samples, and moisture content was analyzed for the dried samples (dried at 60 oC ), it has been added, please see line 319. Moisture values are only used for calculating the dry weight.
40. Line 312: methyl tert-butyl ether (MTBE)
Reply: Thanks for your suggestion. Done, please see line 322.
41. Line 318: EXTRASYNTHASE . Please always confirm what is suggested to you.
Reply: Thanks for your suggestion. Done, please see line 327.
42. Line 320: How did they evaluated HPLC purities? Based on the peak areas, and then subtracting the area of impurities, or these impurities are specified by supplier?
Reply: Thanks for your comment. The HPLC purities were evaluated based on both the peak areas (subtracting the area of impurities) and these impurities specified by supplier.
43. Line 329: “certain amounts of individual compounds (similar for Line 383)”
Reply: Thanks for your suggestion. Done,, please see lines 338 and 394.
44. Line 335: “a series dilutions of β-carotene solutions”
Reply: Thanks for your suggestion. Done, please see lines 344-345.
45. Lien 337: membrane material is important here, PTFE, PVDF, CE, etc.
Reply: Thanks for your comment.The membrane material is nylon. It is added, please see line 346.
46. Line 371: what is sample 5? If it is the sample number from a Table, please refer to the table.
Reply: Thanks for your comment. It is the sample number from Table S7. It is added, please see lines 381-382.
47. Line 382: This section is too long and confusing. It is advised to separate sections as extract preparation, optimization and analysis as done in carotenoid section.
Reply: Thanks for your comment. Done, please see lines 393, 398 and 410.
48. Line 371-392: Please rewrite this paragraph (6 lines for one sentence) with short and clear sentences. Please do not use unnecessary parenthesis. Which method were used for further optimization and optimized extraction?
Reply: Thanks for your comment. This paragraph has been rewritten. Please see lines 399-401.
49. Line 406: “Analysis of Polysaccharide contents.” Plural since other headings are also plural and you have many samples.
Reply: Thanks for your suggestion. Done, please see line 415.
50. Line 424: Antioxidant activity or capacity? Please be consistent. “Antioxidant activity assays.” (Plural). Please specify which extracts were used for antioxidant activity assays here. “Extract I and Extract II as prepared in section X and X, respectively, were used for all antioxidant activity assays”
Reply: Thanks for your comments. This title has been revised as “Antioxidant Activity Assays”, please see line 433. The sentence“Extract I and Extract II as prepared in section 3.3.2 and 3.4.2, respectively, were used for all antioxidant activity assays” has been added in lines 444-445.
51. Line 425: “Brand-Williams“ Please check the manuscript for typing errors.
Reply: Thanks for your suggestions. Done, please see line 434. The manuscript for typing errors have been checked.
This manuscript is a resubmission of an earlier submission. The following is a list of the peer review reports and author responses from that submission.
Round 1
Reviewer 1 Report
Manuscript molecules-489009, despite its scientific interest, should not be accepted for publication. Authors should rewrite the sections “Results and Discussion” and “Conclusions”, because they have used repeatedly the expressions “have significant difference”, “have no significant difference” and other similar ones, that are incorrect. In my opinion, they should use “are significantly different” or a similar expression. Moreover, other important flaws need correction:
It is necessary that authors clarify if they have tested Daodi herbs. Comments in lines 127-134, 160-170, 180-183, and 201-203 may induce to readers to consider that authors have compared FLB extracts with a Daodi herbs extract.
Lines 82-83: the terms “C30 liquid chromatography” and “has” are inadequate. It should be better “C30 reversed-phase columns used in liquid chromatography” and “gets”.
Lines 123-125: This sentence is difficult to understand. It should be rewritten.
Lines 127-129: What means “to have a better clinical therapy” in relation to Daodi-herbs? This sentence should be rewritten.
Lines 137-138: Please, rewrite the sentence. Flavonoids are a class of phenolic compounds; what means “different concentrations of methanol and ethanol”?
Lines 144-147: Have you measured carotenoids in the ethanol extracts?
Line 177: write “moreover”, not “and”
Lines 215-217: This sentence should be rewritten. It is difficult to understand.
Subsection 2.6: There are hundreds, even thousands of scientific papers that show a high and positive correlation between the content of polyphenols and the antioxidant activity of plant extracts. Authors should give some hypothesis to explain why that correlation has not been observed in extracts of Lycium barbarum fruits.
Figure 3: The caption should be corrected. What mean “C: components A and B”?
Reviewer 2 Report
I have carefully reviewed the manuscript which is about the functional components and antioxidant activity of Lycium barbarum fruits from different regions. The manuscript contain data for 78 batches of fruits from 13 regions of China which provide an overview for the variation of functional components between different regions. As functional components, individual carotenoid composition by HPLC-DAD, and total carotenoid, phenolic, flavonoid and polysaccharide contents as well as antioxidant activities (3 assays) by spectrophotometry were tested. In addition, the manuscript provides information on the optimization of carotenoid (Line 69) and phenolic extractions (Line 136) as well as HPLC method optimization and method validation for carotenoids analysis (Line 81), although the data or experimental design were not presented.
The findings show that functional components of L. barbarumvary depend on the region. In overall, the manuscript provides information that may attract attention of the readers of ‘Molecules’.
On the other hand, the manuscript has several scientific weaknesses, especially on methods section such as the selection of analysis methods for each functional component, inappropriate statistical analysis, lack of enough information on plant material and data presentation, lack of discussion as listed in detail below. Therefore, the manuscript need extensive revision.
· Analysis methods: One of the main weakness of the manuscript is the analysis methods and experimental design. Data on carotenoids are very detailed with extraction and HPLC method optimization for the determination of individual carotenoids, and total carotenoids by spectrophotometric assay. However, for hydrophilic components (phenolic, flavonoids, polysaccharides), only unspecific spectrophotometric methods were used, although individual phenolic composition of this sample has been very well established in literature. These spectrophotometric assays are not specific to these components, only giving total values, and easily react with other plant components to overestimate or underestimate real values. Therefore, the manuscript is not focused nor balanced in terms of methodology, while giving too detailed information on one side (for carotenoids), and very little on other side (for phenolics, flavonoids, polysaccharides). That might be the reason for the scattered PCA figures for total phenolic and polysaccharides without any clear separation between regions, while they are clearly separated for carotenoids (Figure 3).
· Statistical analysis: Second important weakness is about the method used for statistical analysis. Was t-test applied after one way-ANOVA? One of the post-doc tests (such as Tukey or Duncan etc.) should be applied especially for a large set of samples as presented in this manuscript. Since each pair of samples should be tested by t-test method to find which one is different, it would result a large number of tests which is unnecessary as well as prone to give erroneous results due to using the same p-value for each case. Please refer to a statistical analysis book for details on when to apply ANOVA+ post-hoc test or t-test. In Table 1, statistically different samples were marked, however it is not clear which one is different from which sample, and their order. Labeling by letter after post-doc test would be more informative. No statistical analysis was given in Tables S3 and S4 for phenolic and polysaccharide contents. It is impossible to see differences and follow the discussion.
· Methods: More information should be given on plant material collection. What are the differences between different batch of samples collected from the same regions? What are the sample sizes? How many biological replicates did they collect? I suggest the authors to pool data and report only one data for each region unless they want to discussion differences between sample batches. If they want to present values for each sample batch, they have to give more information on each batch and also discuss why this difference is important. Optimisation data were not given. How they decided the optimum conditions. It will be better to give these results as well. It is better to write optimisation parameters in methods section as well.
· Antioxidant activity: Since which extracts (hydrophilic or lipophilic) were used for antioxidant activity assays are not clear from text, it is inconclusive whether antioxidant activities are measured for lipophilic (carotenoids) part or hydrophilic functional components. For this manuscript, it should be measured for both of them.Another problem is that different units were used for different assays. For instance; mmol/ g DM for FRAP and mg/mL for DPPH and ABTS. Carotenoid, phenolic and polysaccharide contents were reported as mg/g DW. These differences might be the real cause of low correlations presented in Table 3.
· Correlation: Unless there is an important finding worth to discuss, this part should be eliminated. Because it does not add value to the manuscript. It is also very well accepted now that antioxidant activity assays react with anything in plant extracts (even vitamin C) and does not need to be correlated with specific components such as phenolics.
· Languages: In overall, the manuscript should be extensively improved for its language as well as the presentation and discussion of the results.
· Title should be updated as …..from different regions in China’.
· Discussion:There is almost no discussion and proper comparison with literature.